# OpenReview forum: "ScoreMix: Synthetic Data Generation by Score Composition in Diffusion Models Improves Recognition"
_ICML.cc/2026/Conference — ICML 2026 regular_

### Official Review · Reviewer_xQzW · 2026-03-03

**Soundness:** 4
**Presentation:** 3
**Significance:** 3
**Originality:** 3
**Overall Recommendation:** 4
**Confidence:** 3

**Summary:**

This paper proposes ScoreMix, a generative data augmentation method designed to mitigate the over-reliance on external foundation models or large-scale datasets in sensitive domains such as face recognition. By performing convex combinations of class-conditional scores along the reverse diffusion trajectory, ScoreMix generates highly challenging augmentation samples. Furthermore, the authors empirically demonstrate a weak correlation between the generator’s condition space and the discriminator’s embedding space. Their analysis reveals that mixing classes that are distant in the discriminator’s embedding space yields the most significant performance gains.

**Compliance With Llm Reviewing Policy:**

Affirmed.

**Final Justification:**

The rebuttal addressed my concerns, keeping the score

**Key Questions For Authors:**

（1）The empirical validation is conducted on WebFace160K, a dataset characterized by relatively balanced class distributions. In practical face recognition deployments, however, data often exhibit pronounced class imbalance. Whether ScoreMix maintains its robustness and efficacy under such skewed distributions remains an open question. Further experimental analysis under imbalanced settings is strongly encouraged to substantiate the claimed advantages.
（2）The font size used in Figure 10 is too small, which significantly hinders legibility.

**Limitations:**

Although this work primarily focuses on applying ScoreMix for data augmentation in face recognition, its generalizability to other vision tasks, such as natural image classification or object detection, remains unsubstantiated.

**Strengths And Weaknesses:**

Strengths: The proposed method effectively translates complex "score space" operations of diffusion models into intuitive augmentation strategies through rigorous mathematical definitions. Notably, the discovery that "mixing distant classes yields superior gains" provides a valuable and non-intuitive contribution to the understanding of generative data augmentation.
Weaknesses: The current schematic of the ScoreMix framework in Figure 1 lacks sufficient detail. The authors are encouraged to either enhance this illustration or provide a comprehensive technical roadmap in the methodology section, thereby elucidating the operational details of ScoreMix.

---

> ### Author Rebuttal · Authors · 2026-03-30
>
> We thank the reviewer for the positive assessment and for the specific suggestions on imbalance and presentation.
>
>
> ---
> ### Q1: robustness under class imbalance
> This is an important point. Our current main experiments use WebFace160K because it provides a balanced and controlled setting for isolating the effect of the augmentation strategy and for comparing fairly against AugGen. We agree that robustness under realistic long-tail imbalance is important and currently underexplored in the submission. We will revise the paper to make this limitation more explicit and add experiments on a curated imbalanced subset.
>
>
> ---
> ### Q2: Figure readability and methodological clarity
> Thank you. We increased font sizes in the affected figures and improved the ScoreMix schematic / methodology by adding a generator block that produced the trajectories so that the operational flow is easier to follow at a glance.
>
>
> ---
> ### Limitation: why face recognition (FR) is the main testbed?
>
> ### **General clarification**:
> We chose FR as the primary benchmark for three reasons.
>
> * FR is a particularly relevant setting for self-contained augmentation: collecting large labeled datasets is difficult due to privacy, legal, and policy constraints. This directly matches the motivation of the paper, namely improving recognition without relying on external datasets or pretrained foundation models. This framing is central to the paper’s setup and claims.
>
> * FR uses strong discriminators with metric-learning objectives such as ArcFace/AdaFace, making improvements more meaningful than in simpler closed-set classification settings. Our method is therefore tested in a demanding regime, rather than on a weak classifier. This is consistent with the paper’s experimental setup and comparisons against IR50/IR101 FR systems across multiple public benchmarks.
>
> * FR offers a broad evaluation protocols, although training is done on WebFace160K, testing is performed on 8 public benchmarks spanning high-quality and harder real-world conditions. This provides stronger evidence than a single closed-set test accuracy on one dataset, e.g., on ImageNet's validation set.
>
> To probe beyond FR, we additionally ran a preliminary experiment on CUB-200, as suggested by Reviewer ZG2j. **Under the same small-generator setup used for our controlled experiments, the generator was unstable and often produced low-quality reproductions/mixes; in this regime**, augmentation did not improve over the discriminative baseline. We sepnd more time on the convergence methodologies for better generation quality and report in the final revision.
>
> | Backbone | Method | Discriminator train data | Generator train data | Test accuracy |
> | :--- | :--- | :--- | :--- | :---: |
> | IR18 | Baseline | CUB200 train | CUB200 train | **0.2580** |
> | IR18 | AugGen Mix | CUB200 train + 100 class-mix classes | CUB200 train | 0.2446 |
> | IR18 | ScoreMix | CUB200 train + 100 class-mix classes | CUB200 train | 0.2304 |
>
> Our takeaway at present, evidence supports ScoreMix most strongly in FR, where both the generatodr and evaluation pipeline are well established.

---

> > ### Author Rebuttal · Reviewer_xQzW · 2026-04-04
> >
> > The rebuttal addressed my concerns, keeping the score

---

> > > ### Author Response · Authors · 2026-04-07
> > >
> > > Thank you to reviwer for acknowledging our rebuttal. To further clarify generalization beyond face recognition, we completed an additional self-contained experiment on ImageNet1K. As in the paper, the setup is fully self-contained: the generator and the discriminative models were trained only on ImageNet1K train, with no external data or pretrained models. Using two standard `timm` backbones and a simple cross-entropy objective, ScoreMix improved Top-1 validation accuracy in both cases (mean ± std over 2 seeds):
> > >
> > > | Method | Generator | Added synthetic classes | Backbone | Top-1 val. acc. |
> > > | :--- | :--- | :---: | :--- | :---: |
> > > | Baseline | — | 0 | ResNet50 (`timm`) | 0.4353 ± 0.0010 |
> > > | ScoreMix | EDM2-L | 490 | ResNet50 (`timm`) | **0.4453 ± 0.0019** |
> > > | Baseline | — | 0 | FastViT-S12 (`timm`) | 0.4507 ± 0.0069 |
> > > | ScoreMix | EDM2-L | 490 | FastViT-S12 (`timm`) | **0.4600 ± 0.0018** |
> > >
> > > These experiments are more limited than our FR evaluation, but they provide preliminary evidence that the method is not specific to FR alone. We will add these experiments to the final version.

---

### Official Review · Reviewer_qTAe · 2026-03-10

**Soundness:** 3
**Presentation:** 3
**Significance:** 3
**Originality:** 3
**Overall Recommendation:** 4
**Confidence:** 3

**Summary:**

The paper proposes a method for synthetically augmenting a dataset of face images. This is done through mixing class-conditioned scores along reverse diffusion trajectories. In addition to just combining multiple images for augmentation the approach allows for proportional mixing of them. The approach requires no external dataset or model and instead trains a diffusion model from scratch on the training dataset. Experiments reported in the paper demonstrate that this approach leads to improved model performance across 8 public face recognition benchmarks. The experimental results are thorough and explore different class selection strategies, different backbones, and combining more than two classes.

**Compliance With Llm Reviewing Policy:**

Affirmed.

**Key Questions For Authors:**

Can this work be used for other visual tasks as well? Is this specific to face recognition or similar methods can be used for other classification tasks?

Does the augmentation need to be redone for every model being trained or is it something that could be precomputed? Can the cost be amortized?

Are all the “mixed” generated faces valid representations of a face? Are there failure modes in generation and does that affect the final results?

**Limitations:**

Yes

**Strengths And Weaknesses:**

Strengths:
- The model does not rely on an external source (data or a model) for data generation, ensuring a fully self-contained system and full control of data provenance without risk of “poisoning” your data sources through external model use
- The paper is clearly written and easy to follow. I especially liked the highlighting of the main takeaways.
- The algorithm is explained clearly and considering that code will be made available should be reproducible
- The experiments cover a number of ablations: choosing which classes to mix; using more than two classes for mixing; using learnable vs fixed discriminators
- The experiments are performed on 8 benchmark datasets demonstrating the generalizability of the proposed approach

Weaknesses:
- Can this work be used for other visual tasks as well? Is this specific to face recognition or similar methods can be used for other classification tasks?
- The proposed augmentation increases the computational cost of training a face recognition system and increases the complexity of the whole training pipeline.
- Are all the “mixed” generated faces valid representations of a face? Are there failure modes in generation and does that affect the final results?

---

> ### Author Rebuttal · Authors · 2026-03-30
>
> We thank the reviewer for the positive assessment and for emphasizing reproducibility and the breadth of the ablations.
>
> ----
> ### Q1: applicability beyond face recognition
> At present, our evidence is strongest for FR.
> Please see the Limitation in response to Reviewer xQzW (this is done for space constraints).
> We ran a preliminary CUB-200 experiment and found that the same small controlled generator setup was unstable and did not improve the baseline. We therefore do not claim validated generalization beyond FR in the current version instead, we will revise the discussion to present this as an important direction for future work rather than an established result. We mainly chose FR as the primary benchmark for three reasons.
>
> * FR is a particularly relevant setting for self-contained augmentation: collecting large labeled datasets is difficult due to privacy, legal, and policy constraints. This directly matches the motivation of the paper, namely improving recognition without relying on external datasets or pretrained foundation models. This framing is central to the paper’s setup and claims.
>
> * FR uses strong discriminators with metric-learning objectives such as ArcFace/AdaFace, making improvements more meaningful than in simpler closed-set classification settings. Our method is therefore tested in a demanding regime, rather than on a weak classifier. This is consistent with the paper’s experimental setup and comparisons against IR50/IR101 FR systems across multiple public benchmarks.
>
> * FR offers a broad evaluation protocols, although training is done on WebFace160K, testing is performed on 8 public benchmarks spanning high-quality and harder real-world conditions. This provides stronger evidence than a single closed-set test accuracy on one dataset, e.g., on ImageNet's validation set.
>
> To probe beyond FR, we additionally ran a preliminary experiment on CUB-200, as suggested by Reviewer ZG2j. **Under the same small-generator setup used for our controlled experiments, the generator was unstable and often produced low-quality reproductions/mixes** in this regime, augmentation did not improve over the discriminative baseline. We sepnd more time on the convergence methodologies for better generation quality and report in the final revision.
>
> | Backbone | Method | Discriminator train data | Generator train data | Test accuracy |
> | :--- | :--- | :--- | :--- | :---: |
> | IR18 | Baseline | CUB200 train | CUB200 train | **0.2580** |
> | IR18 | AugGen Mix | CUB200 train + 100 class-mix classes | CUB200 train | 0.2446 |
> | IR18 | ScoreMix | CUB200 train + 100 class-mix classes | CUB200 train | 0.2304 |
>
> ---
> ### Q2: does augmentation need to be redone for each model?
> Not necessarily. The synthetic data can be precomputed once and reused. In addition, our cross-backbone results show that the selected class pairs transfer well across recognition architectures, which suggests that the most expensive selection/generation steps need not be repeated from scratch for every downstream model. We simply re-do them in our experiments for controlled experimentation.
>
> ---
> ### Q3: are all mixed faces valid? are there failure modes?
> For 2-plet mixing after generator convergence, we did not observe non-face or obviously meaningless samples in our runs after generator convergence, the typical failure mode was instead reduced utility, not invalidity. In contrast, when going beyond 2-plet mixing we did observe noticeably more overlaps/disfigurements, which is consistent with the weaker downstream performance of m>2 mixing. We will add this clarification and additional qualitative examples.

---

> > ### Author Rebuttal · Reviewer_qTAe · 2026-04-01
> >
> > The rebuttal addressed my concerns, keeping the score and suggest accepting this paper.

---

> > > ### Author Response · Authors · 2026-04-07
> > >
> > > We thank the reviewer for acknowledging our rebuttal. To further clarify applicability beyond face recognition, we completed an additional self-contained experiment on ImageNet1K, matching the paper’s setting: the generator and the discriminative models were trained only on ImageNet1K train, with no external data or pretrained models. Using two standard `timm` backbones and a simple cross-entropy objective, ScoreMix improved Top-1 validation accuracy in both cases (mean ± std over 2 seeds):
> > >
> > > | Method | Generator | Added synthetic classes | Backbone | Top-1 val. acc. |
> > > | :--- | :--- | :---: | :--- | :---: |
> > > | Baseline | — | 0 | ResNet50 (`timm`) | 0.4353 ± 0.0010 |
> > > | ScoreMix | EDM2-L | 490 | ResNet50 (`timm`) | **0.4453 ± 0.0019** |
> > > | Baseline | — | 0 | FastViT-S12 (`timm`) | 0.4507 ± 0.0069 |
> > > | ScoreMix | EDM2-L | 490 | FastViT-S12 (`timm`) | **0.4600 ± 0.0018** |
> > >
> > > While preliminary relative to our main FR study, these results suggest that ScoreMix can also improve a non-FR large-scale classification setting. We will add these experiments to the final version.

---

### Official Review · Reviewer_NPAz · 2026-03-13

**Soundness:** 3
**Presentation:** 3
**Significance:** 3
**Originality:** 3
**Overall Recommendation:** 4
**Confidence:** 4

**Summary:**

The paper introduces a novel, self-contained synthetic data augmentation method designed to enhance the performance of discriminative models (face recognition moodels here). The core mechanism, ScoreMix, generates "hard" synthetic samples by applying a convex combination (α+β=1) of class-conditioned scores during the reverse diffusion process. This approach ensures that generated samples remain on-manifold by preserving the expected score magnitude and directional integrity learned by the diffusion model. In a word, ScoreMix provides a practical and robust framework for maximizing recognition performance using only a single available dataset.

**Compliance With Llm Reviewing Policy:**

Affirmed.

**Final Justification:**

I think this is a good paper and i suggest to accept this paper.

**Key Questions For Authors:**

1. Why do the generator’s condition space and the discriminator’s embedding space exhibit such weak correlation under standard alignment metrics like CKA, despite being trained on the identical dataset

2. Why does forcing the generator's outputs to align with the discriminator’s class centers (via alignment regularization) actually decrease recognition performance？it is better to have more discussions and insights here.

3. Any solution to reducde the computational overhead.

**Limitations:**

As stated in weakness.

**Strengths And Weaknesses:**

Pros:
1. The method is independent of external foundation models commercial APIs, or third-party datasets. Both the generator and the initial discriminator are trained from scratch on the same target dataset.

2. Mathematical Rigor through Convex Combinations.

3. Across eight public face recognition benchmarks, ScoreMix improved accuracy by up to 7%, which are significant improvement.

Cons:

1. ScoreMix is more computationally expensive than single-condition generation; generating m-plet samples requires roughly m times the cost of previous methods (like AugGen), which may limit its scalability for extremely large-scale data augmentation

2. The effectiveness of the method currently peaks at mixing two classes (m=2); empirical tests show that mixing three or more classes fails to match the performance gains of simple pair-wise mixing using current diffusion generators

---

> ### Author Rebuttal · Authors · 2026-03-30
>
> We thank the reviewer for the positive evaluation and for highlighting both the practical significance and the computational cost.
>
> ---
> ### Q1: why are generator condition space and discriminator embedding space weakly correlated?
> Our interpretation is that the two spaces are optimized for different objectives. The discriminator embedding space is explicitly trained to separate identities under a recognition loss, whereas the generator condition space is optimized only insofar as it helps denoising and image reconstruction. As a result, the two spaces need not preserve the same pairwise geometry, even when trained on the same dataset. Empirically, this explains why condition-space distance is a weak selector while discriminator-embedding distance is much more informative for selecting useful pairs.
>
> ---
> ### Q2: why does alignment regularization reduce recognition performance?
> Our experiments suggest a diversity–coherence trade-off. The alignment regularization moves generated samples closer to class centers, producing more coherent but less diverse samples. For recognition training, however, diversity within an identity is crucial. over concentrating samples around the center reduces useful intra-class variation and therefore hurts downstream recognition, even if the generated images appear more class consistent. This is the main reason we do not advocate this regularization for augmentation.
>
> ---
> ### Q3: reducing computational overhead
> The extra cost is concentrated in synthetic sample generation. Two practical mitigations are possible.
>
> 1. This cost can be amortized once generated, the synthetic set can be reused for training multiple discriminators, and our cross-backbone experiments already indicate substantial transfer of pair selection. This is consistent with the paper’s computational framing and transfer results. Additionally, the upfront compute cost of generation is offset during inference: a smaller model trained with our augmentation surpasses the performance of a larger model trained without it, resulting in long-term computational savings during deployment.
> 2. sampling can be partially cached/precomputed. In particular, for frequently reused class pairs or fixed seeds, one can save reverse diffusion trajectories or generated samples offline and reuse them across runs. We will clarify this practical point in the revision.

---

> > ### Author Rebuttal · Reviewer_NPAz · 2026-04-01
> >
> > The authors have addressed my concerns. I will keep my score and suggest to accept this paper.

---

### Official Review · Reviewer_ZG2j · 2026-03-13

**Soundness:** 3
**Presentation:** 2
**Significance:** 3
**Originality:** 3
**Overall Recommendation:** 4
**Confidence:** 3

**Summary:**

This paper introduces ScoreMix, a data augmentation strategy for face recognition that operates without external models or datasets. A class-conditional diffusion generator and a discriminator (IR50 with ArcFace) are both trained on the same dataset (WebFace160K, ~160K images, 10K identities). Synthetic images are then produced by linearly combining score predictions from two different identity conditions during reverse diffusion ($S_{\text{mix}} = \alpha \cdot S_A + \beta \cdot S_B$ with $\alpha + \beta = 1$). These synthetic images are added to the training set to improve downstream recognition.

Beyond the generation method, the paper conducts a systematic investigation into which class pairs should be mixed. The main empirical finding is that selecting pairs distant in the discriminator's embedding space yields noticeably larger gains than selecting based on the generator's learned condition space (Diff = 2.52 vs. 0.11--0.56 in Table 3). The paper further examines the geometric relationship between these two spaces via CKA/CKNNA, proposes a theoretical bound on order preservation across architectures (Theorem G.1), and shows that pair selection transfers across backbone models (Table 2). Experiments span 8 public face recognition benchmarks, where ScoreMix consistently outperforms training on original data alone and surpasses a larger IR101 backbone.

**Compliance With Llm Reviewing Policy:**

Affirmed.

**Final Justification:**

The rebuttal resolved my two score-driving concerns.
M1 (label protocol): confirmed as new synthetic class per
pair (10K+10K=20K). M2 (non-mixed control): three-setting
experiment with variance shows ScoreMix outperforms
reproductions on most metrics, supporting the mixing
contribution. M3 (single domain) remains — CUB-200 was
negative, though the ImageNet1K follow-up shows preliminary
positive evidence. The rebuttal reinforced my prior
assessment. I maintain 4 (Weak Accept).

**Key Questions For Authors:**

**Q1 (Label assignment — response would likely change my evaluation).** Please confirm: does each $(c_A, c_B)$ pair define a new synthetic identity class in the augmented training set, resulting in $10\text{K original} + 10\text{K synthetic} = 20\text{K}$ total classes? If a different labeling scheme is used, please describe it. This is inferable from Appendix K but should be stated explicitly. A clear answer would resolve M1 and increase my confidence in soundness.

**Q2 (Non-mixed reproduction control — response would strengthen the paper).** Can you provide the result for $\mathcal{D}_{\text{orig}}$ (0.16M) + 0.2M same-class reproductions (single-condition generation, no mixing)? The Close Embedding Cosine result in Table 3 suggests this would show only modest gains, but an explicit number would directly decompose the contribution of mixing vs. data volume.

**Q3 (Variance — helpful for calibration).** Can you report mean $\pm$ std over 3 seeds for: (a) the WebFace160K baseline (Avg in Table 3), (b) Dist Embedding Cosine (Avg in Table 3), and (c) ScoreMix vs. AugGen in Table 1? Given the variance shown in Table 8 ($\pm 2.20$ for one setting), this would clarify which differences are robust.

**Q4 (Beyond face recognition — desirable but not blocking).** Have you attempted ScoreMix on any non-face fine-grained recognition task? Even a small experiment on CUB-200 or Stanford Cars would substantially broaden the significance. If not feasible, a discussion of what properties of FR make ScoreMix especially suitable would be informative.

**Limitations:**

Partially. The authors acknowledge the higher computational sampling cost relative to AugGen and note that the method has been validated only on face recognition (Section 5). The Impact Statement identifies deepfake misuse as a potential risk, and the paper is commendably transparent about several negative results. One point worth clarifying: the Abstract's "no information leakage" framing suggests a privacy guarantee, but what is actually demonstrated is independence from external data — these are different properties, and the wording should be adjusted to avoid overclaiming. Authors should be rewarded for being upfront about limitations.

**Strengths And Weaknesses:**

**Strengths**

**S1 (Soundness — Controlled experimental design).** The class-selection experiment in Table 3 is the highlight of this paper. All 12 strategies operate under matched conditions: the same 0.2M synthetic budget, the same generator, the same sampler, the same number of denoising steps. The only variable is how class pairs are chosen. Under this setup, Dist Embedding Cosine achieves Avg 67.44 while Close Embedding Cosine reaches only 64.92 — a gap of 2.52 that cannot be explained by differences in data volume or generation quality. This is a clean result that directly supports the paper's central claim about the importance of selection strategy.

**S2 (Soundness — Breadth of ablations and negative results).** The paper presents five negative or null findings: (i) condition-space distance is uninformative for selection (Diff = 0.11--0.56 in Table 3); (ii) freezing discriminative features as generator conditions causes training divergence (Section 4.3); (iii) alignment regularization improves fidelity but hurts diversity and downstream performance (Section 4.4, Figures 5--6); (iv) 3-way mixing does not outperform 2-way (Table 3); (v) additional sampler steps beyond 32 yield no consistent benefit (Table 7). Reporting these failures maps the design space and saves future researchers from unproductive directions.

**S3 (Significance — Augmentation vs. scaling).** Table 1 shows that IR50 trained with ScoreMix augmentation outperforms IR101 trained on original data across all benchmarks, while maintaining IR50's lower inference cost. The compute breakdown in Table 8 makes this concrete: augmented IR50 training takes 4.10h vs. 5.60h for IR101 on original data. This is a useful finding for practitioners deciding between bigger models and better data.

**S4 (Originality — Class-selection framework).** While score composition in diffusion models is established (Liu et al., 2022; Bradley et al., 2025), and self-contained FR augmentation via condition mixing exists (AugGen; Rahimi et al., 2025), the systematic investigation of embedding-distance vs. condition-distance selection with controlled comparisons is new. The discovery that these two spaces are weakly correlated despite being trained on the same data (Figure 4, Appendix E) is a genuinely interesting observation.

**S5 (Soundness — Cross-backbone robustness).** Table 2 demonstrates that pairs selected by AdaFace-IR101 transfer to ArcFace-IR50 with most of the gain preserved (+2.76 vs. +3.22 for self-selection). The full CKA matrix in Table 5 across 10 configurations, with RandN as a sanity check, provides solid evidence that the selection strategy is not backbone-specific.

**S6 (Presentation — Convex combination motivation).** Figure 2 provides intuitive visual evidence for why $\alpha + \beta = 1$ matters: off-diagonal cells show clear degradation. The three-part justification in Section 3.1 (score magnitude preservation, manifold interpolation, projective composition) effectively combines intuition with theoretical grounding.

---

**Weaknesses**

*Major (would meaningfully affect my evaluation if unresolved):*

**M1 (Presentation — Label assignment not in main text).** The paper does not explicitly state in the main body how ScoreMix samples are labeled during discriminator training. In a margin-loss pipeline like ArcFace, whether a mixed sample is assigned to a new synthetic class or to one of the source identities determines whether a new class center is created or an existing one is shifted — these have different training dynamics. From Appendix K ("mixed identity," "synthetic folder"), the arithmetic ($10\text{K pairs} \times 20 = 0.2\text{M}$), and the AugGen precedent, I infer that each pair defines a new synthetic identity class. But this should not require inference — it is a core aspect of the method and deserves one clear sentence in Section 3.2 or 4.1. I note this as a presentation issue rather than a soundness flaw, since the information is recoverable from the appendix and predecessor work.

**M2 (Soundness — No explicit non-mixed reproduction baseline).** The paper does not include a "$\mathcal{D}_{\text{orig}}$ + same-class reproductions" control, which would isolate the contribution of inter-class mixing from additional data volume. Table 3 partially addresses this: Close Embedding Cosine — mixing the most similar identity pairs — gains only +1.10 over baseline, while Dist Embedding gains +3.62, under identical budgets. If gains were driven purely by data volume, this gap should not exist. I find this indirect evidence reasonably convincing, though an explicit non-mixed control would have been stronger.

**M3 (Significance — Single-domain evaluation).** All experiments use WebFace160K for face recognition. The paper claims broader applicability to "other structured-input recognition domains" (Section 5) but provides no supporting experiments. Even a small-scale experiment on one additional fine-grained recognition task would have meaningfully broadened the paper's significance for a general ML venue.

---

*Minor (do not individually affect my recommendation):*

**m1 (Soundness — No multi-seed variance).** Tables 1 and 3 report single-run numbers. Table 8 shows non-trivial training variance ($32.63 \pm 2.20$ for one setting). Some reported differences are large enough to be robust (e.g., C-1e-6 in Table 3: 78.62 vs. 71.86), but others are small (Avg-H in Table 1: 93.87 vs. 93.78). Reporting mean $\pm$ std over 3 seeds for key comparisons would strengthen the claims.

**m2 (Soundness — Theorem assumptions).** Theorem G.1 assumes isotropic Gaussian misalignment on the orthogonal complement with energy matching ($\sigma^2 = (1-\rho^2)/(N-1)$). Table 10 shows the bound predicts overlap to within 5--10\%, which is encouraging at the order-of-magnitude level but does not validate the distributional assumption itself. Since the practical takeaway (higher CKA $\to$ better transfer) is already demonstrated in Table 2, the theorem's contribution is more conceptual than predictive.

**m3 (Presentation — Notation overloading).** $S$ denotes both the denoiser network (Eq. 2: $S_\theta$) and score outputs (Eq. 4: $S_A$, $S_B$). $\theta$ is acknowledged as shared between discriminator and denoiser (end of Section 3) but used without consistent subscripting thereafter. Separating these (e.g., $\epsilon_\theta$ for the network, $\mathbf{s}$ for score outputs) would reduce ambiguity.

**m4 (Presentation — Main text vs. appendix balance).** The CKA theorem occupies considerable main-text space (Section 4.3, Eq. 6) despite limited predictive value beyond Table 2. Meanwhile, the hard-sample characterization (Appendix K, Table 9: distance to sources $0.687 \pm 0.040$ vs. non-sources $0.775 \pm 0.005$), the alignment loss formulation (Appendix D), and sensitivity studies (Appendix I) are deferred. Promoting Table 9 into the main text would support the "hard on-manifold samples" claim more directly than the theorem.

**m5 (Presentation — AugGen comparison).** Table 1 reports "AugGen" (B-1e-6: 29.40) and "AugGen Repro" (B-1e-6: 15.71) with a large gap. The text attributes this to "performance inconsistencies previously reported" but does not clarify which numbers come from the original paper vs. the authors' reproduction, making it hard to identify the fair comparison point.

**m6 (Presentation — "No information leakage" wording).** The Abstract's claim of "no information leakage" is somewhat misleading. What the paper actually demonstrates is independence from external data sources, which is a meaningful and well-motivated property. However, this is distinct from guaranteeing that the generator does not memorize training samples. Adjusting this wording to more precisely describe what is achieved (e.g., "no external data dependency") would improve accuracy.

---

> ### Author Rebuttal · Authors · 2026-03-30
>
> We thank the reviewer for the careful reading and detailed feedback. We are glad that the the reviwer find our controlled class-selection study, the negative results, and the cross-backbone robustness found valuable.
>
> ---
> ### Q1 / M1: label assignment for ScoreMix samples
> Yes. Each mixed pair defines a new synthetic identity class during discriminator training. For example, if the original training set has $l$ classes and we add one synthetic class per selected pairs ($l_{mix}$ classes), the classifier is trained with $l + l_{mix}$ classes. We agree this should have been stated explicitly in the main text, and we will add one clear sentence to Section 3.2 / 4.1.
>
> ---
> ### Q2 / M2: non-mixed reproduction control
> We agree that an explicit control is important. We therefore ran the requested experiment by adding non-mixed reproductions and comparing three settings: (i) reproductions treated as the same original class, (ii) reproductions treated as new classes, and (iii) dummy new centers without training samples.
>
> | Setting | Head/Backbone | #Classes | B-1e-06 | B-1e-05 | C-1e-06 | C-1e-05 | TR5 |
> | :--- | :--- | :---: | :---: | :---: | :---: | :---: | :---: |
> | Baseline | ArcFace-IR50 | 10K | 33.17±0.49 | 72.50±0.41 | 70.29±0.64 | 78.57±0.37 | 66.60±0.28 |
> | Repro as dummy centers | ArcFace-IR50 | 10K + 30K | 34.31±0.23 | 72.26±0.06 | 70.38±0.20 | 78.53±0.28 | 66.50±0.11 |
> | Repro as new class | ArcFace-IR50 | 10K + 10K | 34.60±0.84 | 72.99±0.06 | 72.84±0.10 | 80.17±0.07 | 66.85±0.06 |
> | Repro as same class | ArcFace-IR50 | 10K | 33.36±0.72 | 77.64±0.18 | 76.18±0.09 | 83.17±0.16 | 66.55±0.21 |
> | AugGen mix | ArcFace-IR50 | 10K + 10K | 34.83±0.39 | 76.21±0.07 | 75.02±0.48 | 82.91±0.49 | 66.60±0.24 |
> | ScoreMix | ArcFace-IR50 | 10K + 10K | 35.98±0.59 | 77.36±1.10 | 77.07±1.41 | 84.10±0.72 | 67.99±0.28 |
>
> These results support the reviewer’s interpretation: improvement is not explained by synthetic data volume alone. Non-mixed reproductions help, but ScoreMix remains best overall, indicating that inter-class mixing contributes beyond simple reproduction, but the fact that adding reproduction helps sets was interesting, we will add a scaled version of this experiment to the final version.
>
> ---
> ### Q3 / m1: variance across seeds
> We agree that reporting variance is important. Due to rebuttal-time constraints, we completed repeated runs first for the new reproduction-control table above and now report mean+-std there. We are continuing the repeated runs for the main comparisons and will update the key tables accordingly in the revision.
>
> ---
> ### Q4 / M3: beyond face recognition
> **Please see the Limitation in response to Reviewer xQzW (this is done for space constraints)**. We added a preliminary CUB-200 experiment. The result is currently negative because the generator was unstable under the same small controlled setup, so we do not use it to claim broad generality. Instead, we will revise the paper to state more clearly that the current evidence is strongest for FR.
>
> ---
> ### m2: theoretical bound and practical value
> We agree that the theorem is more conceptual than predictive. Our goal was not to claim a tight predictive bound, but to provide a theoretical explanation for why pair ordering can remain stable under partial geometry preservation across backbones/loss-heads. The main practical evidence remains empirical, especially the cross-backbone transfer results. We revised the text to make this positioning clearer.
>
> ---
> ### m3: notation overloading
> Agreed. We will separate the denoiser network notation from its score outputs, e.g., using a dedicated network symbol such as $g_{\theta}$ for the network and distinct notation for the outputs.
>
> ---
> ### m4: appendix vs. main text
> Agreed. We moved the more relevant experimental evidence into the main text, in particular the clarification around reproductions and the hard-sample characterization, and compress the theorem discussion.
>
> ---
> ### m5: AugGen comparison
> Agreed. We explicitly marked which AugGen numbers are taken from the original paper and which are our own reproductions, so the comparison point is unambiguous, we used a * in the footnote of the table.
>
> ---
> ### m6: "no information leakage"
> Agreed. We replaced this wording with the more precise claim that our method has no external data/model dependency. The current wording is stronger than what is actually established by the paper.

---

> > ### Author Rebuttal · Reviewer_ZG2j · 2026-04-03
> >
> > The authors addressed my two score-driving concerns (M1, M2)
> > directly and substantively.
> >
> > M1 (label assignment): Confirmed — each mixed pair defines a new
> > synthetic identity class (10K original + 10K synthetic = 20K).
> > This resolves the presentation ambiguity I raised. The planned
> > addition to Section 3.2/4.1 is appropriate.
> >
> > M2 (non-mixed reproduction control): The authors provided a
> > three-setting control experiment (same-class, new-class, dummy
> > centers) with multi-seed variance, which goes beyond what I
> > requested. ScoreMix outperforms non-mixed reproductions on most
> > metrics, supporting the claim that inter-class mixing contributes
> > beyond simple data-volume increase. I note that "Repro as same
> > class" is competitive on B-1e-5 (77.64 vs 77.36), which is an
> > interesting finding that further highlights how label assignment
> > strategy affects training dynamics.
> >
> > M3 (beyond FR): The CUB-200 result is negative, and the authors
> > appropriately revised their claims to be FR-specific rather than
> > overclaiming generality. This limitation remains but was already
> > reflected in my original score.
> >
> > Minor points (m2--m6) were all accepted with concrete revision
> > plans.
> >
> > I maintain my score of 4 (Weak Accept). The rebuttal reinforced
> > my original assessment.

---

> > > ### Author Response · Authors · 2026-04-07
> > >
> > > We thank the reviewer again for the careful follow-up. To address the remaining limitation on generality beyond face recognition, we completed an additional self-contained experiment on ImageNet1K. As in the paper, the setup is fully self-contained: the generator and the discriminative models were trained only on ImageNet1K train, with no external data or pretrained models. Using two standard `timm` backbones and a simple cross-entropy objective, ScoreMix improved Top-1 validation accuracy in both cases (mean ± std over 2 seeds):
> > >
> > > | Method | Generator | Added synthetic classes | Backbone | Top-1 val. acc. |
> > > | :--- | :--- | :---: | :--- | :---: |
> > > | Baseline | — | 0 | ResNet50 (`timm`) | 0.4353 ± 0.0010 |
> > > | ScoreMix | EDM2-L | 490 | ResNet50 (`timm`) | **0.4453 ± 0.0019** |
> > > | Baseline | — | 0 | FastViT-S12 (`timm`) | 0.4507 ± 0.0069 |
> > > | ScoreMix | EDM2-L | 490 | FastViT-S12 (`timm`) | **0.4600 ± 0.0018** |
> > >
> > > These results are more limited than our FR study, but they provide preliminary evidence that ScoreMix is not restricted to FR alone. We will add these experiments to the final version and keep the claims appropriately scoped.

---

### Decision · Program_Chairs · 2026-04-30

**Decision:**

Accept (regular)

**Comment:**

This paper focuses on privacy- and policy-constrained synthetic augmentation. The paper proposes ScoreMix, a synthetic augmentation method for face recognition that mixes class-conditioned diffusion scores to generate hard manifold samples without relying on external data or pretrained models. Four experts in the field reviewed this paper, all rating it as weak accept. Reviewers agreed that the paper is clearly written and organized, technically sound, clearly motivated, and supported by controlled experiments, particularly the class-selection study showing that mixing identities distant in the discriminator embedding space provides the strongest improvements. The main strengths are the self-contained augmentation setup, the empirical improvements across several benchmarks, and the analysis of condition-space versus embedding-space geometry. The main weaknesses are the limited support for ScoreMix beyond face recognition, the computational overhead, and some unclear details (e.g., labeling protocol, baselines, and figures). Based on the reviewers’ feedback and the authors’ satisfactory rebuttal, which addresses the most important concerns, I recommend it for acceptance. However, the reviewers have raised some issues that should be addressed in the final camera-ready version of the paper.